# Is That “Mr.” or “Ms.” Lemon? An Investigation of Grammatical and Semantic Gender on the Perception of Household Odorants

**DOI:** 10.3390/brainsci12101313

**Published:** 2022-09-28

**Authors:** Theresa L. White, Caitlin M. Cunningham, Mary L. Zampini

**Affiliations:** 1Department of Psychology, Le Moyne College, Syracuse, NY 13214, USA; 2Department of Mathematics and Statistics, Le Moyne College, Syracuse, NY 13214, USA; 3Department of World Languages, Literatures, and Cultures, Le Moyne College, Syracuse, NY 13214, USA

**Keywords:** olfaction, language, household odors, grammatical gender, anthropomorphized gender

## Abstract

Linguistic relativism is the idea that the structure of language influences thought. The present study investigates linguistic relativism by asking whether people who speak a gendered language think of objects in a way that is consistent with the grammatical gender (more masculine or feminine) more than people who speak a language without that linguistic feature. Odorants are unique stimuli for examining this question, because they carry a semantic gender without a biological one, yet gender is thought to be a central dimension of their perception. Canadian participants in the present study (native French or native English speakers) described a set of labeled odorants that varied systematically in grammatical and semantic gender to produce an implicit gender measure and also explicitly rated them for gender. Analysis of the implicit femininity contained in participants’ descriptions showed a tendency for both native French and native English speakers to produce descriptions that were consistent with semantic gender; moreover, there were no effects of grammatical gender on implicit or explicit ratings. These results suggest that in a bilingual culture, French speakers may have been influenced by the anthropomorphism associated with odorants rather than grammatical gender.

## 1. Introduction

When people encounter a household odor outside of the context provided by everyday life, they are easily able to comment on the scent’s pleasantness but find it remarkably difficult to produce the name of the smell [1,2,3]. The difficulty that people have in correctly associating odors with their names is well documented (e.g., [1]) and when unassisted, people are typically only successful in naming half of the odorants in a given set of smells [1,4]. Though people can give the same odor the same name on repeated presentations, the name may differ from the veridical label [5]. In English, descriptions of odors are often quite abstract and unrelated to odors [6] possibly reflecting a separate, odor-specific lexicon that is limited [7,8]. Reasons proposed for this odor/language difficulty are varied, with some suggesting that an inherent variability of the stimulus contributes to the problem [1,9]; however, odors are named much better in a multiple-choice situation [10], indicating that innate variability may not be the sole source of the difficulty. Evidence from patients with aphasia suggests that the inferior frontal gyrus appears to be involved in odor naming [11,12,13]; information is relatively unprocessed at the temporal pole, providing limited information to language centers even for neurologically intact individuals. While competition for cortical processing time and space could contribute to the weakened connection between odors and language [14], other evidence [15] suggests that the difficulty may have less to do with brain organization than with cultural differences. Many studies of odorant identification involve participants who are WEIRD (white, educated, industrialized, rich, democratic); other cultures, such as the Jahai people of the Malay Peninsula seem to be better at attaching language to odors than WEIRD populations [8]. The culture may also form a context for the ability to name odors successfully [16]. Thus, although there is debate as to whether the lack of ability to reliably and accurately identify odors [17,18] arises due to cortical organization [11,12,13] or cultural differences [7,8], it is clear that as a result of the difficulty, the perception of olfactory stimuli is vulnerable to the top-down influences of language. For example, knowing the identity of an odor source alters its ratings of intensity, pleasantness and familiarity [19,20]. In addition, participants rate the pleasantness of odorants in accordance with the connotation of an accompanying label [21,22]. Thus, the connection between odorants and language is complex, yielding both strong and weak effects on perception and cognition. What are the limits of language’s influence on olfactory thoughts, and what aspects of language are important to olfactory perception? The current study will address this question by examining the influence of grammatical gender and semantic gender on olfactory perception by speakers of French and English.

### 1.1. Linguistic Relativism

The extent to which language modifies the way we think about the world is a long-standing question in language research. Perhaps of most interest is whether knowledge of a *particular* language influences thought, because if this is true, then speakers of different languages may have perceptions of the world that are shaped specifically by the language that they speak. The idea that knowledge of a particular language shapes thought is encompassed in the Sapir-Whorf Hypothesis [23,24]; linguistic relativism is the version of this hypothesis that claims that knowledge of a particular language can influence and shape thought (e.g., [25,26]). Studies that have addressed the Sapir-Whorf Hypothesis have produced decidedly mixed results (e.g., [27,28]). Still, the phenomenon has been widely researched and generally supported ([29], p. 498), though much is still unknown regarding the nature and strength of the interaction between language and thought.

### 1.2. Linguistic Relativism and Grammatical Gender

One of the more commonly studied linguistic constructs in relation to linguistic relativism is grammatical gender, especially as it relates to how speakers perceive inanimate objects. Grammatical gender is a widespread linguistic property of nouns that does not indicate sex or gender per se, but instead denotes formal agreement between nouns and other words that are syntactically related to them, such as adjectives and verbs [30]. Languages may vary according to the number of grammatical gender classes that they have: for example, Spanish and French have two genders—masculine and feminine—while other languages such as German have three genders—masculine, feminine, and neuter. In these languages, the grammatical gender of inanimate nouns is largely arbitrary, and a noun that is classified as masculine in one language may be feminine in another (e.g., the word for *milk* is masculine in French, *le lait,* but feminine in Spanish, *la leche*). Other languages, like English, do not have grammatical gender, and although there may be some masculine/feminine word pairs that can be used to refer to male and female human beings separately (e.g., *actor* vs. *actress* in English), nouns are not typically assigned gender. However, even languages without grammatical gender may exhibit a tendency to anthropomorphize some objects by assigning a semantic gender; for example, in English, a ship may be referred to as a “she”. The typological variation with respect to grammatical gender in language raises an interesting question for linguistic relativism: Do people who speak languages that employ grammatical gender (such as French) think of objects as more masculine or feminine than people who speak languages that do not have this linguistic feature (such as English)?

A number of studies (e.g., [27,30]) have examined the extent to which grammatical gender may influence the way speakers of gendered languages perceive objects; that is, whether or not the grammatical gender of a noun influences their perceptions of the (personified) properties of that noun. Boroditsky, Schmidt, and Phillips [25], for example, summarized a series of studies that examined the role that grammatical gender may play in German and Spanish speakers’ perceptions of objects. Critically, the studies were conducted in English, a language without grammatical gender. In one of the studies, German and Spanish speakers were given a list of 24 inanimate objects in English and asked to describe each object by writing down the first three adjectives that came to their mind that described the object. The objects differed with respect to their grammatical gender in the participants’ native language. The researchers found that the participants generated more masculine adjectives for those objects that were grammatically masculine in their native language and more feminine adjectives to those objects that were grammatically feminine in their native language. To illustrate, the word *key* is masculine in German but feminine in Spanish; German-speaking participants generated adjectives such as *hard*, *heavy*, *jagged*, and *metal* for this word, whereas Spanish speakers generated adjectives such as *golden*, *intricate*, *tiny*, and *lovely*. Although this interesting finding is often referenced, it was only reported as part of a chapter, and quantitative data were not reported; a later study provided a more detailed account of these findings and the technique itself [31]. Nevertheless, Boroditsky et al. [25] argue that studies such as these support linguistic relativism and show that the way in which speakers think about objects is influenced by the grammar of the language that they speak.

Other studies, however, suggest that there are limitations to the role of grammatical gender on the perception of objects (e.g., [32,33]). For example, in a study by Boutonnet, Athanasopoulos, and Thierry [34], the results of an English-language matching task showed no effect of grammatical gender on explicit task performance in bilingual Spanish speakers, though Event-Related Potentials (which are implicit) were influenced by gender inconsistency. Other studies [35,36] have found effects of grammatical gender for some stimuli (names of animals or objects), but not others (pictures of animals or objects). In addition, at least one publication has reported the failure to replicate the experiment by Boroditsky, Schmidt, and Phillips [25] in two different studies [37], suggesting that the result may not be particularly robust. This sentiment has been echoed in the results of a recent meta-analysis [26], which concluded that many of the results supporting linguistic relativism with regard to grammatical gender seemed to be task-specific and subject to a number of possible alternative explanations.

Many studies of linguistic relativism (e.g., [25]) do not address cultural influences, which may be a confounding variable because these influences co-vary with language. The role of culture is an important one in influencing language, and thus, linguistic relativism [38,39]. Thus, it is possible that observed gender effects are due, at least in part, to associations of objects with gender that are influenced by culture independently of grammatical structure. One such cultural influence is the way in which objects without biological sex may nevertheless have an anthropomorphized, or semantic, gender conferred upon them through the use of a gendered pronoun [40], even in languages that do not employ grammatical gender [41]. In fact, some authors have argued that grammatical gender is simply a form of personification that may reflect different attributes of objects [42,43] or the sex of the person most associated with the object [44].

### 1.3. Olfactory Perception and Grammatical Gender

Although a number of studies have examined speakers’ perceptions of physical objects, very few have considered whether bilingual speakers tested in English describe non-spatial objects such as odorants in a way that reflects the grammatical gender of their native language (similar to the findings of Boroditsky et al. [25] for physical objects), whether the gender specifier acts as a prime for the concepts of masculinity or femininity [45], or whether such speakers personify the odors in the same way as others in their culture. People are easily able to classify odorants by gender [46,47]. Zellner et al. [47] showed that the color that people chose to correspond with a fragrance was related to the perceived masculinity/femininity of that perfume-and that labeling a unisex perfume as male or female caused differences in the colors that were selected to go with it. Fiore [48] showed that when a hypothetical person was theoretically wearing a feminine floral perfume, participants rated the person as having fewer traditional male characteristics. In short, these two reports show that thinking of a perfume as male or female seemed to also bring a corresponding stereotypical concept to mind; there is limited evidence to suggest that household odorants may be able to elicit the same gender stereotypes. Kerr, Rosero, and Doty [46] found that a hypothetical person was described as more masculine or feminine depending on the smell of their home (like onion or lemon), with femininity associated with increased perception of hygiene and pleasantness. Hovis, Sheehe, and White [49] also showed that the level of masculinity associated with some household odors could alter whether a gender-neutral description of a person that was associated with that odor was perceived as male or female.

Masculinity and femininity are thought by perfumers to be central dimensions of olfactory perception [50]. Since molecules clearly have no natural gender, this raises the question of how people make gender determinations when classifying odors as masculine or feminine. That is, do the physical aspects of an odorant that are associated with stereotypical concepts of masculinity or femininity drive this decision? Or might the grammatical structure of language influence whether an odorant is considered masculine or feminine? More specifically, do people who speak languages that employ grammatical gender think of odorants in a way that is more consistent with their grammatical gender than people who speak languages that do not?

In one of the few studies to directly address the potential influence of grammatical gender on olfactory perception, Speed and Majid [51,52] asked French and German speakers to read descriptions of fragrances, then smell perfumes marketed as either masculine or feminine and rate each one on different dimensions. Each fragrance was presented twice: once with a description that included ingredients that had feminine grammatical gender and once with a description that included ingredients of masculine grammatical gender. Afterward, participants completed an odor recognition task, in which they smelled several odors and indicated whether or not each odor was one that they had smelled in the earlier phase of the experiment. Results showed that the participants remembered the fragrances better when the grammatical gender of the ingredients used in the description matched the marketed gender of the fragrance. In discussing their results, Speed and Majid [51] proposed that grammatical gender information from the descriptions could combine with the gender of the fragrance to strengthen memories when the two types of information are congruent with respect to gender. Their findings also support proposals (e.g., as in [21]) that odor perceptions are influenced by both verbal context and sensory experience.

### 1.4. Linguistic Relativism, Cultural Anthropomorphism, and Olfaction

As seen from the preceding review, research studies have shown evidence for the following: Grammatical gender may influence speakers’ perceptions of objects; the concept of gender is prompted by smell, at least with respect to perfume; and semantic gender, as reflected in the personification of objects, may be influenced by culture. The current study brings these ideas together and extends the literature in significant ways with two main objectives: To evaluate the level of consistency in the semantic gender of everyday odorants, and secondly to assess the level of influence of native language on thinking about odors. The central question asks if the presence of grammatical gender influences the hedonic ratings of common household odorants. Because gender has been described as a central dimension of odorants [50], they are unique stimuli for examining this question by carrying a semantic gender without a biological one. Further, odorants are paradoxically both difficult to label and highly influenced by labels, so the influence of language on these stimuli is worthy of exploration. To address this central question, we asked whether people who speak a language that employs grammatical gender (French) think of some odors as more masculine or feminine than do people who speak a language without grammatical gender (English). Comparing participants who are tested in different languages is intrinsically different, as there is no way to know that the instructions and stimuli are truly the same in both languages; to control for this, all participants in the present study were tested in English. In an effort to reduce inter-participant variability due to the known difficulty in generating a name for odorants, the English name for each odorant was provided to the participants. In order to help control for culturally determined semantic influences, all participants were from Montreal, Canada. The participants were asked to (i) write down adjectives that were descriptive of a set of labeled odorants that varied systematically in grammatical gender and in anthropomorphized semantic gender (as an implicit measure of gender) and (ii) explicitly rate the masculinity and femininity of the odorants. The implicit response measure was included because some studies (e.g., [34]), show different results between implicit and explicit measures. The current study thus extends the work by Boroditsky, Schmidt and Phillips [25] on the influence of grammatical gender on speakers’ perceptions of physical inanimate objects to odors. It also extends work by Speed and Majid [51,52] by looking at household / food odorants, rather than fragrances, and considers the semantic gender of such odorants. Finally, it extends the work on culture and language [38,53] by examining the perceptions of odorants by groups of people that share a geographically defined culture (Montreal, Canada) but differ with respect to language (English vs. French).

## 2. Materials and Methods

### 2.1. Participants

Thirty-two participants (20 women, 12 men) were recruited from the greater Montreal area through postings at the Montreal Neurological Institute and Hospital, which is associated with McGill University. Participants were healthy volunteers. Exclusion criteria were multiple chemical sensitivity, history of neurological or psychiatric disease, any other condition leading to impaired sense of smell, or allergies to odors used in this study. All participants signed an informed consent form presented in their native language that had been approved by the Montreal Neurological Institute and Hospital Research Ethics Board prior to beginning the experiment.

The participants were specifically recruited such that their native (first) and primary language was either French (16 people; *M*_age_ = 22.81, *SD* = 6.50) or English (16 people; *M*_age_ = 22.56, *SD* = 4.00). That said, all participants also had at least some second language (L2) experience. All but one of the native French speakers had English as their second language and were French-English bilinguals, although they considered French to be their primary language. The other native French speaker had Spanish as their second language. The native English speakers had a wider range of L2s and L2 abilities. In order to confirm participants’ degree of L2 experience, all participants completed the Language Experience and Proficiency Questionnaire (LEAP-Q; see [54]) as part of the experiment. The LEAP-Q is a questionnaire of self-reported proficiency, for which respondents answer a series of questions on an 11-point Likert scale (0–10). Scores of 7 or 8 and higher indicate good proficiency in a second language, while scores of 3 or 4 and lower indicate fair to low proficiency. In many studies, speakers with scores of 7 or 8 and higher are considered bilingual, while those with scores of 3 or 4 and below are considered to be effectively monolingual ([55], pp. 946–947). Details of the second language experience of participants are shown in Table 1. Based on the numbers reported in Table 1, five of 16 native-English speaking participants may be considered bilingual, while seven of the 16 native French-speaking participants may be considered bilingual.

### 2.2. Materials

The 16 odorants for this study were similar to the ones used by Wright, White, and Zampini [56]. A list of these odorants can be found in Table 2, along with their manufacturers, their semantic genders (produced by native English speakers and reported by [56]) and French grammatical gender (as found in [57]). The odorants were selected such that eight of them carry the same semantic gender in English as the grammatical gender in French (four masculine, four feminine), while the other eight odors diverge in this regard (four masculine, four feminine). The set of odorants was contained in opaque brown 30 mL vials that had been correctly labeled with the English name of the odorant.

Two questionnaires were also used for this study: the LEAP-Q (described above) and a scaling booklet. The scaling booklet contained four pages per odorant, each with a separate scale: Two visual analogue scales (ratings of masculinity and femininity) and two variants of the gLMS [58,59], one [58] for intensity and one [59] for pleasantness ratings.

### 2.3. Design

This experiment was a 2 (Speaker’s Native Language-English and French) × 2 (French Grammatical Gender of Odorant–masculine or feminine) × 2 (Semantic Gender of Odorant-masculine or feminine) design. The dependent variables were either taken directly (explicit ratings of pleasantness, masculinity, femininity, intensity) or assessed indirectly through the independent assessment of responses by other individuals (ratings-see Section 2.3.2
*Analysis of Adjectives* below). The final dataset contains 512 total rating sets, with 32 subjects rating 16 odors each. This repeated measures design was accounted for using a mixed methods approach.

#### 2.3.1. Procedure

Each participant was briefly interviewed in order to exclude participants who would react adversely to smells. Each successfully screened participant was tested individually in a quiet room in English so as to avoid leading the subject by using grammatical gender (after [25]), and participants were not fully informed of the study’s purpose until the completion of testing. Participants completed an implicit evaluation of the odors, followed by an explicit evaluation of the odors, and an evaluation of language experience.

In the implicit test of gender, each participant was presented with each of the odorants listed in Table 2. The odorants were presented to the participant in a random order, one at a time, by the experimenter. Each participant was asked to provide descriptors for the odorants and to avoid giving a description that simply entailed using the name on the bottle’s label (noun stem) with the suffix “−y”. Participants were instructed as follows:

“I’m going to give you 16 different odors, one at a time. I would like for you to write down the first three adjectives (not names) that come to mind to describe each odor. An *adjective* is a ‘describing’ word, not a name. So, if I were to give 3 adjectives describing this testing table, I might say, “Light, shaky, and mobile”. Using words like “table-like” or “table-y” would not be acceptable as an adjective. So, I want you to use this booklet to write down the first 3 adjectives that come to mind about each odor, and to use a different page for each odor.”

After all stimuli were described, participants smelled each odor a second time (in a new random order) and explicitly rated each odorant using the scaling booklet for pleasantness, femininity, masculinity, and intensity (in that order for each odorant). Following these explicit ratings, participants were asked to complete the LEAP-Q. Native French participants were also asked to translate each English odor name into French to make certain that their usage of grammatical gender did not vary due to differences in dialect.

#### 2.3.2. Analysis of Adjectives

After the completion of the study, the adjectives generated in the implicit test were arranged in alphabetical order and presented individually to five native English speakers (3 women, 2 men; *M*_age_ = 20.8, *SD* = 1.48) who were unaware of the circumstances in which the words were collected. These native English speakers determined whether the adjective represented a masculine or feminine descriptor (after [25]).

An examination of the reliability of the raters with Fleiss’ Kappa revealed no evidence that any specific rater was significantly different from the others (kappa score = 0.312, as per [60]). Thus, the percentage of raters who classified each adjective as feminine was averaged across the three descriptions produced in response to each odorant; this measure was considered indicative of *implicit femininity* for each odorant and is reported below as the percent femininity score.

## 3. Results

### 3.1. Implicit Ratings Based on Odorant Descriptions

Based on previous research (e.g., [25]), it was hypothesized that odors that were mismatched in terms of semantic gender and French grammatical gender would be given adjectives by native French speakers that were more consistent with the grammatical gender than those given by native English speakers. In other words, the adjectives produced by the native French participants would be less feminine for those words with masculine grammatical gender and more feminine for those words with feminine grammatical gender when compared to the adjectives given by the native English speakers.

Even though they were explicitly told not to do it, participants tended to create adjectives by adding a “-y” suffix to an odor source name (e.g., “minty”). This is common in untrained individuals [61], however, and these adjectives were not removed from the present data set.

### 3.2. Analysis of the Primary Hypothesis

First, in order to analyze the hypothesized effect, it was necessary to ensure that the native French participants knew the (intended) grammatical gender of each odorant. Therefore, at the end of the testing session, each native French participant was asked to translate the English names for the odorants into French. In a few instances, the translation varied from the odorant name intended; that is, the grammatical gender of the French translation provided by the participants matched the intended name 97% of the time. In the formal analysis, each odorant was classified into the grammatical gender based on the French translation produced by the speakers themselves.

With that in mind, the results for the implicit test, in which the participants generated adjectives to describe each odor, will be presented first. As indicated above, a percent femininity score was calculated for each adjective based on how the adjective was rated by an independent group of English speakers. Overall, average implicit femininity scores for the native French speakers proved to be lower than the native English speakers (French: *M* = 52.96, *SD = 21.86*; English: *M* = 57.56, *SD* = 20.5), indicating that the native French speakers were less likely to describe an odorant using feminine terms. Figure 1 shows this in more detail, illustrating the average implicit femininity of the odorants in each of the combinations of semantic and French grammatical gender.

Initial examination of Figure 1 shows that generally, odorants that had both a semantic and grammatical gender that were feminine led participants to produce adjectives that implicitly reflected femininity more than masculinity. Likewise, odorants that had both a semantic and grammatical gender that was masculine led people to produce fewer feminine adjectives. A comparison of the two odorant groups that share the same semantic gender in Figure 1 suggests that the native French speakers may have a tendency to provide more adjectives that coincide with grammatical gender of the odorant as revealed by slightly lower implicit femininity scores for feminine odorants with masculine grammatical gender as compared to feminine odorants with feminine grammatical gender and slightly higher implicit femininity scores for masculine odorants with feminine grammatical gender as compared to masculine odorants with masculine grammatical gender. However, the difference is very small, and native English speakers show a similar trend. Finally, an examination of the responses to the odorants that were “mismatched” as to their semantic and grammatical gender in Figure 1 shows that on the whole, participants produced adjectives that were consistent with the semantic gender, regardless of whether their native language was French or English.

This hypothesis was formally explored using a linear mixed model, with random effects for the repeated speakers and odors, and with main fixed effects of native language spoken, semantic gender, French grammatical gender, and their interactions. Means and standard deviations for each dependent variable by speaker group for each odorant is shown in Table 3. Each term was added to the model sequentially and tested for significance using a likelihood ratio test. Gender of the subjects was considered as a possible covariate, but average implicit femininity score was not significantly different between genders, even after accounting for language group and the semantic gender of the odors. As such, it was not included in the final model. The difference in mean average percent femininity between the native French and native English speakers was significant (χ^2^(1) = 5.81, *p* = 0.016). Semantic gender also strongly and significantly predicted average percent femininity (χ^2^(1) = 23.9, *p <* 0.001); its interaction with language, however, was not significant (*p* = 0.29), indicating that the effect of semantic gender was the same for participants of both language groups. Adding French grammatical gender to the model either as a main effect (*p* = 0.76) or as an interaction (*p* = 0.80) with semantic gender was not statistically significant. That is, for the native French speakers, the grammatical gender was not statistically significant in predicting the average percent femininity of the adjectives used to describe the odors. Thus, implicit femininity was not affected by grammatical gender, and the initial hypothesis is not confirmed. Before moving on to the explicit ratings, it should be noted that the language classifications included in the analysis detailed above were made on the basis of what each participant reported as their primary language. Restricting the analysis to only those English speakers without any experience with a gendered second language as reported on the LEAP-Q was not possible due to the paucity of data since only four participants met this criterion. In addition, an analysis with a leave-one-out cross validation confirmed that there were no influential participants as the results did not significantly change with the removal of any one participant.

### 3.3. Explicit Ratings as a Secondary Test of the Primary Hypothesis

In addition to the implicit test just described, participants were asked to assign explicit ratings of masculinity, femininity, pleasantness, and intensity to each odorant. Figure 2 shows the mean ratings (and standard errors) of each of these four explicitly measured attributes. For ease of viewing, the ratings were normalized to a 0–100 scale. As no differences between native French and native English speakers emerged with these explicit ratings, Figure 2 collapses across participants’ native language. One can easily observe from the figure that odorants selected a priori via pilot experiment as having a feminine semantic gender are rated as more feminine than masculine, while those selected as having a masculine semantic gender are rated as more masculine than feminine. This manipulation check also indicates that participants were able to assign gender to household odorants, but notice that the masculine odors are not rated as particularly (above 50) masculine or feminine.

It is also worth noting that in 13% of the trials, participants gave 0s when asked to rate the femininity or masculinity of the given odor. This may indicate an unwillingness to assign gender to odors, rather than a rating of 0 femininity or masculinity for the specific odor. It could also indicate a tendency on the part of the participants to view masculinity and femininity as a binary characteristic (an odor is either masculine or feminine) rather than as a continuous scale.

Next, to determine the extent to which the implicit measures of femininity (the percent femininity scores of the adjectives) relate to explicit measures, the percent femininity scores of the adjectives were examined in relationship to each of the explicit measures (pleasantness, intensity, explicit femininity, and explicit masculinity) for both participants groups combined. A linear mixed model with random effects for the repeated odors and subjects, and with fixed effects for pleasantness and intensity was executed for explicit femininity, explicit masculinity, and implicit femininity. The fixed effects were assessed for significance by adding them to a restricted model and calculating likelihood ratio tests.

The mixed model regression with implicit femininity as the response and with pleasantness and intensity as explanatory variables shows a statistically significant positive relationship with pleasantness (β = 0.19, χ^2^(1) = 55.14, *p* < 0.001) and a statistically significant negative relationship with intensity (β = −0.14, χ^2^(1) = 8.82, *p* = 0.003). Including explicit femininity and masculinity in the regression finds a statistically significant relationship with explicit femininity (β = 0.15, χ^2^(1) = 17.17, *p* < 0.001) but not with masculinity. The stronger relationship, however, is with the odor properties (pleasantness and intensity), rather than explicit femininity. There is a significant main effect for semantic gender when it is included in the regression (β = −14.82, χ^2^(1) = 33.8, *p* < 0.001), such that odors identified as semantically feminine a priori received more feminine adjectives.

## 4. Discussion

To summarize, the present study asked questions about the way in which a grammatical feature of language (grammatical gender) and a culturally influenced aspect of language (semantic gender) affect cognition associated with household odorants. Participants who shared a similar cultural background (Montreal, Canada) but differed with respect to the presence or absence of grammatical gender in their native / dominant language (French vs. English) generated adjectives to describe a series of odorants that were either matched or mismatched for French grammatical gender and semantic gender; they also provided explicit ratings of each odorant for femininity, masculinity, pleasantness, and intensity. Overall, the data revealed a strong effect of semantic gender, but little effect of grammatical gender. Average implicit femininity scores for the native French speakers were lower overall than for the native English speakers, but the pattern of results was the same for both speaker groups. That is, although the native French speakers were less likely than the native English speakers to describe an odorant using feminine terms, the grammatical gender of the odorant names in French did not influence their responses. Instead, participants of both language groups showed an influence of semantic gender on their implicit ratings. More specifically, when the odorants were mismatched with regard to grammatical and semantic gender, participants of both speaker groups produced adjectives that corresponded more closely to the odorant’s semantic gender, rather than grammatical gender. In the explicit ratings of the household odorants, participants’ ratings largely aligned with the odorants’ semantic gender as well. That is, odorants with a feminine semantic gender were rated as more feminine, and odorants with a masculine semantic gender were rated as more masculine. In addition, an examination of the relationships between the implicit and explicit measures revealed that percent femininity scores of the adjectives used to describe the odorants showed a positive relationship with the ratings for both pleasantness and explicit femininity and a negative relationship with the ratings for intensity.

### 4.1. Grammatical Gender Did Not Influence the Description of Household Odorants

As mentioned above, analysis of the implicit femininity scores revealed that both speaker groups had significantly higher implicit femininity scores for semantically feminine odorants than for semantically masculine odorants and that grammatical gender, on the other hand, did not significantly impact the responses. These results suggest that in a bilingual culture, native French speakers may have been influenced by the anthropomorphism associated with odorants. Thus, the present data suggests that the grammatical gender of household odorants does not bias descriptions of those odorants toward one gender or the other.

The results of the present study are therefore inconsistent with studies (e.g., [25,41]) showing that speakers of a gendered language think of an object in a way that is consistent with its grammatical gender. However, as discussed above, evidence to this end is divided. Some research suggests that the influence of grammatical gender is constrained to specific situations, such as when verbalization occurs [62] or in response to specific semantic categories, such as animate objects [35,63], neither of which were a part of the present experiment. In addition, the present results using household odorants are in contrast to research that showed that the memory of fine fragrances was improved by gender congruency between a fragrance and its experimenter-provided descriptors [52]; however, the cognitive tasks differ considerably between the two studies. Further, fine fragrances are much more ambiguous and conceptually abstract than the more concrete household odorants of the present research, for which an odorant source is more easily identified [64]. It is possible that the difference between the present findings and those in the literature were due to the concreteness of the stimulus concept, as language is purported to have the most influence on perception when perception is uncertain (e.g., [17]). Odorants that are more easily recognizable and categorizable (such as household odorants) may be more influenced by semantic and cultural aspects of language [53] and less prone to influence from linguistic features like grammatical gender than odorants that are more ambiguous or complex (such as perfumes).

The failure to find an effect of grammatical gender in the current experiment may also have been due, at least in part, to the fact that English was the language used with all participants throughout the experimental protocol. Both languages of a bilingual individual may be activated during the performance of linguistic tasks [65], and it is unknown how much one language affects the other when they are both activated. In the present study, the language in which the experiment was conducted may have formed a relevant context for thought [66]; in other words, the use of English for testing the native French participants may have obscured differences that might have been produced as a result of grammatical gender. Although testing bilingual speakers in English is an extremely conservative test of the hypothesis that the grammatical features of the native language can influence thought, since the influence must be so pervasive so as to appear in a second language, it may also have had the unintended effect of cueing the gender concepts associated with a non-gendered language. Recall, however, that Boroditsky, Schmidt and Phillips [25] found an effect of grammatical gender for their native German and Spanish participants, even though the experiment was conducted in English. More broadly, much psycholinguistics research has shown influence from one language on another in a variety of cognitive tasks performed by bilingual speakers, even when they are actively using only one language (e.g., [65,67,68,69]). Thus, it was reasonable to hypothesize that grammatical gender effects may have been present in the current study, as well, even though it was conducted in English, rather than French.

As a final possible explanation for the discrepancy between the present results and contradictory findings in the literature, all of the speakers in the present study, regardless of their linguistic aptitudes, lived in the Montreal area, and thus shared some level of culture with each other. It is possible that the effect of culture was strong enough to hinder any possible effects of grammatical gender. Other studies have shown that although grammatical gender does have an effect on cognition, the effect is quite small compared to the influence of culture (e.g., [38]). The present findings are consistent with studies that have demonstrated similarity in gender perception of objects between speakers of English and German ([32], pp. 130–141), speakers of German and Italian ([70], as cited by [32]), and speakers of English and German [33], despite differences in grammatical gender. These results lead to the idea that the concepts of masculinity and femininity are based on shared human experiences that may be cultural in nature (e.g., [38,39]), rather than being defined by linguistic features, such as grammatical gender. This idea will be examined more thoroughly in the next section.

### 4.2. Cultural Gender Personification of Household Odorants Is Possible

The participants’ explicit ratings of femininity and masculinity for the household odorants provide further support for the influence of semantic gender on perception. In addition, semantically feminine odors received higher ratings for pleasantness, which is a dimension associated with feminine body odors [71]; likewise, the semantically feminine odors had lower ratings for intensity, a dimension on which higher levels are associated with masculine body odors [72]. Household odorants are unlike biologically based odorants in that their origins are unrelated to people; these smells originate from other objects in nature, such as food or flowers. There is no biological basis of sex for household odorants, yet the present findings support the idea that people can readily assign a gender to household odorants [46], much in the same way that perfumes [47,50] can be personified as having a gender. These results support initial findings by Kerr, Rosero, and Doty [46], who first demonstrated cultural personification of household odors for a smaller set of odorants-only four- and three of these appear in this set of smells. In that study, people sniffed each odor and then described a hypothetical person whose house smelled in that way; some odors caused the hypothetical person to be rated as more masculine or feminine. The present findings show in a larger set that like fine fragrances, household odorants can be associated with a shared cultural stereotype of gender [19] and thus imbued with their own personified gender.

In the present study, participants consistently chose words for specific odorants that reflected gender stereotypical language. In North American culture, the stereotypical woman is gentle, tactful, religious, dependent, interested in self-appearance, and needs security, while the stereotypical man is aggressive, unemotional, ambitious, objective, self-confident, independent, and dominant [73]; these stereotypes are reflected in the language that participants used to describe the household odorants in the present study. For example, they used words such as “airy” or “beautiful” to describe odors that they also explicitly rated as very feminine, while they used words such as “aggressive” or “bold” to refer to odorants that they rated as highly masculine. Previous research [19,48] has argued these cultural stereotypes may be reflected in the gender perception associated with odorants.

Semantic gender is likely driven by associations that people have, both to the odorants themselves and to the constructs accessed by the odorants and the labels [17,22,38]. A bias toward femininity was observed with this set of household odorants, in contrast to the masculinity bias that has been observed with body odorants [74]. One of the reasons that participants did not seem to rate any of these odorants as very high in masculinity may be because the 16 odors in this study were derived from natural objects, which are more associated with femininity (per [75] as cited by [51]). A well-defined understanding of odorant masculinity seemed to be lacking in this study, suggesting that it may not be as relevant as the construct of femininity for olfactory stimuli. This finding generally agrees with the work of Lindqvist [76], which calls into question the concept of odorant-sex associations.

Because participants were able to regularly respond to semantic gender independent of their native language, gender may be one of the dimensions along which odorants may be successfully described [77]; however, it’s important to note that the participants of the current study were frequently (16%) unwilling to assign a gender to a given odorant when asked explicitly. This finding suggests that the assignment of a gender role is not a typical aspect of the perception of household odorants and is in contrast to previous findings in perfumes that the construct of gender is a central aspect of odorant description [50,76].

### 4.3. Limitations

The present study employed both implicit and explicit measures of an odorant’s perceived gender. Construct validity of the measure of implicit femininity was demonstrated by its positive relationship with explicit femininity. However, explicit ratings of masculinity were not inversely related to implicit femininity. This likely underscores the differences in the way that the two tasks assessed gender. In the ratings of the adjectives that contributed to the implicit measure, gender was assumed to be binary, either male or female, while ratings of male and female were able to vary independently in the explicit measure. Future research could evaluate implicit femininity on a continuous variable scale, in order to more directly compare with the explicit measure.

It could be argued that explicitly labeling odors in the present study with their names could lead participants to strategically use linguistic information to make their odor judgments; thus participants may have made their responses based on the labels alone, rather than directly considering the odorants, which may be considered as a type of demand bias [78]. It is more likely, however, that responses reflect an integration of stored knowledge about the label with an olfactory percept. In most experimental situations, especially those that require the performance of a difficult task, participants tend to use all of the information that is given to them. We have no reason to suggest that they would have ignored the olfactory information in order to expedite their own performance. Future research in this area, however, could test bilingual speakers in both labeled and unlabeled conditions to examine more thoroughly the potential influence of labels on odor judgments.

Finally, the present study was conducted entirely in English, a language without grammatical gender. In addition, asking the native French speakers to generate descriptions of the odorants in their second language may have restricted the word pool from which names could be drawn for the native French speakers in the current study. The dependent variable of implicit femininity in the present study was obtained from an evaluation of the adjectives that participants produced to describe the labeled odorants. The task of generating descriptions of odorants at all is extremely difficult, given the poverty of odor source independent words [79], and the addition of the need to search for such descriptors in a non-native language may have decreased the number of available words. Future research, therefore, might examine native French speakers in French and English to see if the language used in the research design triggers differing results with respect to the generation of adjectives and the potential influence of grammatical gender on implicit and/or explicit measures of olfactory perception.

## 5. Conclusions

Semantic gender seems to influence descriptions of household odorants more than grammatical gender. When native French participants were confronted with odorants that had names with grammatical genders that did not match their semantic gender in the present study, there was no evidence that the grammatical gender affected their response. The present results underscore the importance of culture in determining the way that we think about odorants. Just as culture appears to influence the vocabulary associated with odorants, it also seems to influence how we think about the dimension of semantic (personified) gender. The culture that we live in, therefore, appears to influence how we think as well as the language that we speak.

## Figures and Tables

**Figure 1 brainsci-12-01313-f001:**
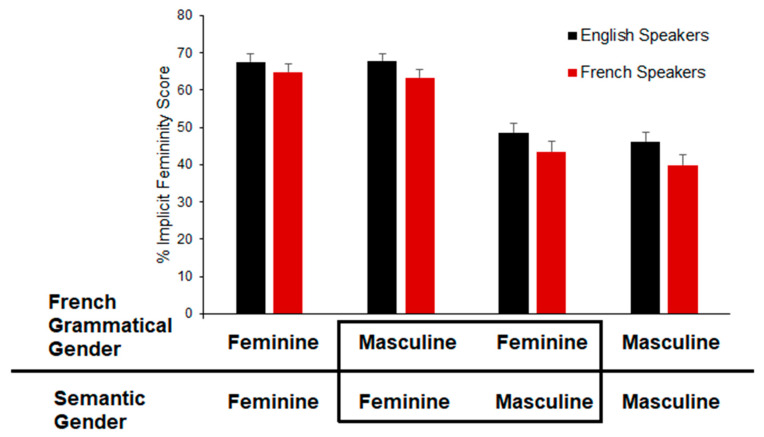
The areas in the box on the x-axis indicate the odorants that are mismatched between semantic gender and French grammatical gender. The bars reflect the mean implicit femininity score, and the error bars represent the standard error.

**Figure 2 brainsci-12-01313-f002:**
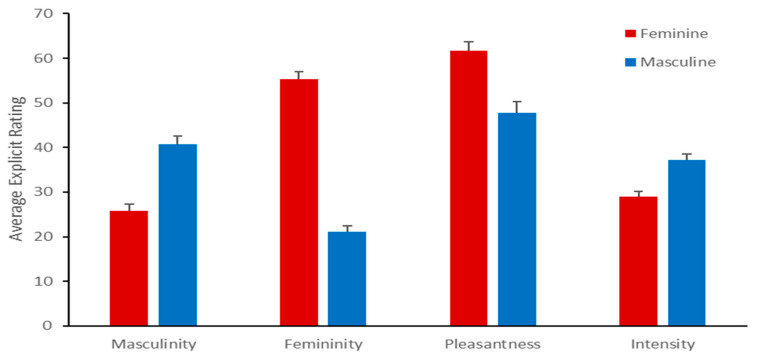
Figure depicts the mean explicit VAS or gLMS ratings of odorants selected a priori to be semantically masculine and feminine, with the error bars showing the standard error of the mean. All ratings were normalized to be on a 0–100 scale (pleasantness was actually rated from −100 to 100).

**Table 1 brainsci-12-01313-t001:** Summary of Participant Group Characteristics.

	French	English
N	16	16
Gender: N Female (%)	12 (75%)	8 (50%)
Age: mean (sd)	22.81 (6.5)	22.56 (4.0)
Leap-Q Results	Second Language: 15 English Proficiency > 7: *n* = 10 Proficiency 5–6: *n* = 3 Proficiency < 4: *n* = 2 1 Spanish Proficiency N/A	Second Language: 9 French: Proficiency > 7: *n* = 1 Proficiency 5–6: *n* = 3 Proficiency < 4: *n =* 5 4 Other (Gendered): Greek: Proficiency 9 Lithuanian: Proficiency 7 Spanish: Proficiency 1 Gujarati: Proficiency 0 3 Other (Non-Gendered): Mandarin: Proficiency 8 Cantonese: Proficiency 6 Japanese: Proficiency 2
N Bilingual (%)	7 (44%)	5 (31.3%)

**Table 2 brainsci-12-01313-t002:** Semantic (anthropomorphized) and Grammatical Genders of Odorants by Language. Masculine gender in both languages is on a gray background for reading clarity. Classification of anthropomorphized gender for native English speakers is based partially on the work of Wright, White, and Zampini [56] and partially on pilot studies. International Flavors and Fragrances (IFF) was the primary manufacturer of the odorants listed in this table.

Odorant	English Semantic Gender	French Grammatical Gender	Manufacturer
Onion	Masculine	Masculine	IFF
Pepper (spice)	Masculine	Masculine	IFF
Peach	Feminine	Feminine	IFF
Banana	Feminine	Feminine	IFF
Peanut	Masculine	Feminine	IFF
Licorice	Masculine	Feminine	IFF
Honey	Feminine	Masculine	IFF
Butter	Feminine	Masculine	IFF
Potato	Masculine	Feminine	PiCs
Beer	Masculine	Feminine	Heineken
Cantaloupe	Feminine	Masculine	IFF
Lemon	Feminine	Masculine	IFF
Pine	Masculine	Masculine	IFF
Garlic	Masculine	Masculine	IFF
Orange	Feminine	Feminine	IFF
Rose	Feminine	Feminine	IFF

**Table 3 brainsci-12-01313-t003:** Means of explicit and implicit ratings by language group. Standard deviations are indicated in parentheses.

	Implicit Femininity		Explicit Pleasantness		Explicit Intensity		Explicit Masculinity		Explicit Femininity	
Name	French	English	French	English	French	English	French	English	French	English
Onion	39.6 (20.4)	37.1 (14.8)	29.8 (19.1)	35.6 (16.4)	57.4 (13.1)	51 (19.4)	17.2 (20.8)	32.2 (27)	6.3 (10)	11.5 (11.9)
Pepper	42.7 (19.3)	53.1 (20.2)	49.4 (16)	57.1 (14)	39.2 (15.5)	34.7 (23)	64.5 (21.6)	39.6 (20.8)	18.5 (21.1)	29.5 (21.9)
Peach	69.3 (12.3)	71.3 (9.7)	74.2 (11.1)	71.5 (14.9)	28.2 (10.9)	34.1 (22.2)	16.4 (15.3)	24.3 (24.5)	78.7 (21.3)	76 (19.8)
Banana	66.2 (12.2)	67.9 (15.2)	66.4 (14.1)	61 (16.4)	24.5 (12.4)	28.9 (23.1)	25.5 (15.3)	31.3 (23.6)	55.9 (17.6)	45.1 (23.8)
Peanut	44.3 (20.9)	46.3 (22.4)	53.9 (10.1)	48.7 (16.6)	33.1 (18.9)	26.2 (13.4)	35.3 (22.5)	32.3 (24.2)	20.6 (16.9)	22.6 (21)
Licorice	41.1 (17.7)	50.4 (16.3)	49.1 (23.4)	55.8 (15.6)	39 (18)	29.8 (15.5)	35.1 (24.5)	38.6 (17.7)	32.3 (30.6)	40.1 (25.9)
Honey	62.7 (17.6)	70.4 (10.9)	58.7 (15.4)	57.6 (11.7)	29.8 (15.4)	30.3 (14)	28.9 (31.8)	31 (18.4)	50.4 (27.6)	48.6 (19.5)
Butter	64 (20.7)	64.2 (14.2)	51.3 (14.7)	56 (18)	19.9 (12.4)	20.3 (10.4)	18.3 (16)	23.6 (19.6)	35.3 (27.7)	32.8 (26.2)
Potato	46.2 (20.7)	49.2 (16.8)	43.2 (14)	42.5 (10.2)	18 (10.1)	31 (20.4)	32.5 (31.8)	39.3 (26.8)	17.2 (21.4)	17.4 (25.3)
Beer	42 (22.2)	48.8 (23.9)	48.7 (21.4)	57.7 (19.9)	36.7 (18.1)	29.1 (15.9)	62.1 (27.2)	63 (20.4)	18.9 (17.7)	18.6 (17.5)
Cantaloupe	69 (10.7)	62.5 (15.8)	58.7 (10.4)	47.7 (12.1)	17 (9.2)	22.1 (15.8)	16.9 (19.4)	18.1 (15.2)	58.3 (17.3)	44.5 (26.3)
Lemon	58.2 (15.2)	74.2 (13.5)	68.9 (13.9)	70.3 (10.4)	30.3 (14.4)	38 (22.4)	35.5 (23)	36.6 (20.4)	51.4 (26.1)	54.6 (20.2)
Pine	48.9 (21.3)	63.3 (11.7)	66.1 (15.1)	63 (14.8)	37.5 (20)	30.4 (18.3)	56.9 (34.9)	54.6 (14.1)	35 (24.4)	32.7 (25.5)
Garlic	29.4 (16.7)	31.3 (16.1)	34.6 (19.3)	30.3 (16.3)	52.3 (19.8)	50.6 (20.5)	24.9 (23.3)	24.3 (27.1)	9.2 (15.2)	8.2 (10.3)
Orange	66.2 (18.9)	73.3 (13.3)	68.2 (14.1)	69.4 (11.1)	25.4 (11.7)	31.4 (16.8)	34.5 (21.9)	38.7 (22.4)	60.3 (19.7)	51.1 (25.7)
Rose	57.3 (25.2)	57.9 (22.5)	53.4 (14.7)	53.9 (15.7)	39.8 (23)	41.8 (23.2)	12.1 (17.3)	19.8 (27.9)	75.1 (30)	67.4 (27)

## Data Availability

Data available upon request to authors.

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
