# Peer review of "Is That “Mr.” or “Ms.” Lemon? An Investigation of Grammatical and Semantic Gender on the Perception of Household Odorants"

_brainsci, 2022, doi:10.3390/brainsci12101313_

Round 1

Reviewer 1 Report

The present study examining the influence of grammatical gender and semantic gender on olfactory perception by speakers of French and English: Do people who speak languages that employ grammatical gender (such as French) think of objects (household odorants) as more masculine or feminine than people who speak languages that do not have this linguistic feature (such as English)?

The paper is well written, the introduction is very clear and covers nicely the literature in the field. The idea to employ household odors as the objects is very exciting. That said I have several concerns related to the small subject groups, their gender and the procedure of odors presentation;

·        *There is relatively small subject number for this kind of experiment- how did the author decided to recruit only 32 participants in total (only 16 in each group). I would recommend recruit more subjects to have at lest 20 in each group.

·        *Moreover, the number of males and females is not equal and it was not specified how many from each gender were in each of the groups (French/English)

·       * Along with that, an important control analysis should test the influence of gender on both implicit and explicit ratings.

·      *  Line 295- “Using these reported numbers to determine “bilingual” status, five of 17 the native-English speaking… “- I am not sure I follow the numbers- at the beginning of the section they reported to have 16 participants in each group and now the authors mention 17?

·       * In addition, the report of the LEAP-Q results is a bit confusing- I would recommend to add a table to present the LEAP-Q results and other characteristics (gender) of each of the groups?

·     *   The authors should justify their decision to show the participants the real label of each odorant during the experiment. I am not sure I understand the rational for this procedure.

·       * It would be informative to have a figure showing all odors and how each odor was rated by each of the groups- for both implicit and explicit tasks.

Author Response

Thank you for taking the time to review our article; we appreciate your kind words and careful reading of the manuscript.  We have endeavored to address your concerns, as well as those of Reviewer 2.  Our responses to your concerns appear below your comments, which are in bold print.

  •       There is a relatively small subject number for this kind of experiment- how did the author decide to recruit only 32 participants in total (only 16 in each group). I would recommend recruiting more subjects to have at least 20 in each group.

Since each subject rated 4 odors within each English Semantic/French Grammatical pairing, the full dataset consists of 512 total observations, with 64 observations within each Language and  English Semantic/French Grammatical pairing group.  These 64 observations are correlated, of course, since they are generated by 16 participants, but this correlation was taken into account in the analysis performed.  The repeated measures aspect of the study design partially accounts for the smaller sample size.  A similar paper by Mickan et al (2014) also used only 15 participants per group.  Although 20 participants per group would indeed be better (as larger sample sizes always are), we cannot recruit more participants in this short time.  We have added a line in the Design section of the manuscript to highlight the total number of odor ratings analyzed, as it is much larger than 16 per group.

  •       Moreover, the number of males and females is not equal and it was not specified how many from each gender were in each of the groups (French/English)

We have added a table with the major characteristics of each group.  There were 8 females and 8 males in the English speaking group and 12 females and 4 males in the French group.

  •       Along with that, an important control analysis should test the influence of gender on both implicit and explicit ratings.

There is no main effect difference in implicit femininity scores between the two genders.  Looking just at average femininity between the two genders, we find no significant difference (p = 0.2073), and adding it as a factor in our full model with language group and English semantic gender is also non-significant (p = .5474).  We added a sentence describing this in the results section of the manuscript.

  •       Line 295- “Using these reported numbers to determine “bilingual” status, five of 17 the native-English speaking… “- I am not sure I follow the numbers- at the beginning of the section they reported to have 16 participants in each group and now the authors mention 17?

       We apologize, but the 17 was a typographical error.  We have 16 participants in each group, and have corrected the text accordingly.

  •      In addition, the report of the LEAP-Q results is a bit confusing- I would recommend to add a table to present the LEAP-Q results and other characteristics (gender) of each of the groups?

We have added the table as requested.

  •      The authors should justify their decision to show the participants the real label of each odorant during the experiment. I am not sure I understand the rational for this procedure.

      We decided to label the odorants in English because the ability to name odorants shows very large variability both within and across participants.  Because we were trying to see if  the name itself influenced odorant perception, people needed to know the identity of the odorant.  We have tried to clarify this idea at the end of the introduction section.

  •      * It would be informative to have a figure showing all odors and how each odor was rated by each of the groups- for both implicit and explicit tasks.

Thank you for your suggestion.  We have added the table as requested.

Reviewer 2 Report

dear authors thank you very much for the paper.

the introduction and discussion sections are extremely long and require some attention to reduce them

please follow mdpi style for referencing, about 5-7 refs are missing from the list please revalidated 

some references are very old and can be omitted during revision.

sample size calculation is required.

sample selection criteria is also needed. did you e.g. screen for certain disorders dyslexia?

detailed description of the administration of methods is needed.

statistical analyses are robust but questionable with a small sample size.

variables need to be described clearly for statistical inference.

I would like to see sensitivity analysis for language differences.

results reported in the tables don't need to be repeated in the text.

discussion need to start with major findings.

I suggest that allsers at a very important paragraph for the practice implications of the study.

one important limitation that might be also reflected in the title that with the small sample size this study is better referred to as a pilot.

I suggest that conclusion is down for language, this is due to the limitation above.

Author Response

Thank you very much for reading our manuscript and for offering suggestions for improving the document.  We have endeavored to address your concerns, as well as those of Reviewer 1.  Our responses to your concerns appear below your comments, which are in bold print.

  •         the introduction and discussion sections are extremely long and require some attention to reduce them. 

The introduction and discussion sections have been streamlined as much as possible, while preserving the thoroughness of the overview of the problem that was noted by reviewer 1.

  •         please follow mdpi style for referencing, about 5-7 refs are missing from the list please revalidated - 

The references have been checked thoroughly, and the document has been changed to mdpi style.

  •         some references are very old and can be omitted during revision.

We appreciate your comment, and having made an effort to streamline the introduction and conclusion, have left only the most relevant references.

  •         sample size calculation is required.

Unfortunately, we do not have an a priori sample size calculation to share.  The original paper by Boroditsky et al (2003) contains no means or standard deviations, and so could not be used to generate an a priori power analysis. The paper by Semenuks et al (2017) has a different measure of gender valence for the adjectives and gives only means, and so also cannot be used. The paper by Mickan et al. (2014) uses the same measure of implicit femininity as the current study, but does not give measures of variation.  Without this information, it is difficult to run a reliable power analysis. This study follows Mickan et al, which had a sample size of 15 in each group, and uses 16 subjects per group.  In addition, this study uses a repeated measures design, so that each subject rated 16 odors, increasing the effective sample size. We have added a line in the design section of the manuscript to highlight the total number of odor ratings analyzed, as it is much larger than 16 per group.

  •         sample selection criteria is also needed. did you e.g. screen for certain disorders dyslexia?

Thank you for noticing this; we did screen for a few health difficulties.  We have clarified this by adding the following into the document in section 2.1, on participants:  “Participants were healthy volunteers. Exclusion criteria were multiple chemical sensitivity, history of neurological or psychiatric disease, any other condition leading to impaired sense of smell, or allergies to odors used in this study.” 

  •     detailed description of the administration of methods is needed. 

We have reviewed the methods section to ensure that the methods are clearly described.

  •     statistical analyses are robust but questionable with a small sample size.

Since each subject rated 4 odors within each English Semantic/French Grammatical pairing, the full dataset consists of 512 total observations, with 64 observations within each Language and  English Semantic/French Grammatical pairing group.  These 64 observations are correlated, of course, since they are generated by 16 participants, but this correlation was taken into account in the analysis performed. The repeated measures aspect of the study design partially accounts for the smaller sample size. 

  •     variables need to be described clearly for statistical inference.

The main variables used in this study are stated in the design section of the manuscript. Further description of those variables is included in the procedures section.

  •     I would like to see sensitivity analysis for language differences.

Leave-one-out cross validation was performed on the main results, where each subject was removed one at a time and the analysis re-run (since we are removing each subject one at a time, and each subject made 16 ratings, this is removing 16 ratings at a time).  The overall difference in implicit femininity scores between French and English remains significant with the removal of a single subject (with all p-values below .03).  The differences in implicit femininity scores due to English Semantic Gender also remains strongly significant (with all p-values below .0001).  The grammatical gender remains not significant in every case.  This indicates that no single subject is influentially driving the results. We included a sentence about this analysis in the manuscript in the results section.

  •     results reported in the tables don't need to be repeated in the text. 

We have read the results carefully and have ensured that there is no unnecessary repetition of the contents of the tables.

  •     discussion need to start with major findings. 

The discussion now begins with a summary of the main findings, and the rest of the discussion has been streamlined to eliminate unnecessary repetition.

  •     I suggest that allsers at a very important paragraph for the practice implications of the study.

We’re sorry, but we don’t understand what the reviewer is asking here.

  •     one important limitation that might be also reflected in the title that with the small sample size this study is better referred to as a pilot.

We respectfully disagree with this suggestion.  This sample size is in line with other published work [see Mickan et al (2014)].

Round 2

Reviewer 1 Report

The authors have addressed my previous comments and questions. The manuscript is improved and I have no further comments. 

Reviewer 2 Report

thank you for addressing my main concerns.